# High Mechanical Properties of AZ91 Mg Alloy Processed by Equal Channel Angular Pressing and Rolling

**Yuchun Yuan [1,\*], Qingfang Guo [1], Jiapeng Sun [1]**  **, Huan Liu [1]** **, Qiong Xu [1], Yuna Wu [1], Dan Song [1], Jinghua Jiang [1] and Aibin Ma [1,2,\*]**

[1] College of Mechanics and Materials, Hohai University, Nanjing 211100, China; gqf770317256@163.com (Q.G.); sun.jiap@gmail.com (J.S.); liuhuanseu@hhu.edu.cn (H.L.); xuqiong@hhu.edu.cn (Q.X.); wuyu_na@126.com (Y.W.); songdancharls@hhu.edu.cn (D.S.); jinghua-jiang@hhu.edu.cn (J.J.)

[2] Suqian Research Institute, Hohai University, Suqian 223800, China

[\*] Correspondence: yychehai@163.com (Y.Y.); aibin-ma@hhu.edu.cn (A.M.); Tel.: + +86-1385-177-2363 (Y.Y.); +86-1391-597-3439 (A.M.)

**Abstract:** Mechanical properties usually take precedence for wrought magnesium alloy when it would be used as a structure material. This paper proposed an approach that achieved high strength in AZ91 Mg alloy. The main procedure combined solution heat treatment, equal channel angular pressing (ECAP), and the subsequent low temperature rolling. After solution heat treatment and ECAP, the alloy had fine grains and excellent ductility, which benefited the following rolling at low temperature. By the following rolling (at 150 °C), the strength was further increased to ~432 MPa with a moderate ductility. This approach was proved effective in refining the grains and accumulating dislocations. The ultrahigh strength was attributed to the high density of dislocations and fine structure. The uniformly distributed fine precipitates also supplied precipitate hardening. Recrystallization that happened during rolling and annealing was the main reason for the moderate ductility.

**Keywords:** magnesium alloy; ECAP; rolling strength; precipitate

## 1. Introduction

Magnesium alloy attracts increasing attention for its low density or high specific strength, which provides considerable weight saving potential in automobile and transport industries [1,2]. Wrought magnesium alloy had higher mechanical properties than the conventional cast magnesium alloy, so it would be the main pattern of magnesium alloys when applied as structural materials [3,4]. However, because of the low ductility and poor workability of magnesium alloys at room temperature, the alloys were generally rolled or extruded at temperatures higher than 200 °C [5,6], which directly led to the big grain size and relatively poor mechanical proprieties, which greatly restricted the industry application of magnesium alloys.

Equal channel angular pressing (ECAP) [7,8] as a normal severe plastic deformation (SPD) procedure have been utilized in magnesium alloys [9–12] in recent years. It refined the grain size of magnesium alloys and improved the strength and ductility simultaneously. Good ductility was achieved in many magnesium alloys by ECAP, but there was still room for the strength to improve [12,13]. The reported tensile yield strengths of Mg–Al series alloys processed by ECAP were typically no more than 250 MPa [9,12,14]. Dynamic recrystallization during ECAP limited the grain refinement and released some dislocations. The specific texture was helpful for the ductility

of the processed alloy, but it also led to a yield strength loss [14,15]. Reducing the ECAP processing temperature [16], or combining the extrusion and ECAP [17] method could produce finer grains and increase the yield strength to ~300 MPa. At our previous study, combining ECAP with solution and post aging rendered higher strength without ductility loss in the AZ91 magnesium alloy, but the yield strength (270 MPa) was still not very high [18].

In this work, in order to obtain a higher strength in AZ91 magnesium alloy, low temperature rolling was implemented following solution heat treatment and ECAP. The pre-solution and ECAP procedure was hoped to refine the grains and improve the formability of the alloy in the followed rolling.

## 2. Experimental Procedure

The commercial AZ91 Mg alloy ingot with a composition of 9.13 wt. % Al, 0.69 wt. % Zn and remainder of Mg was used in this work. The as-cast AZ91 alloy was firstly cut into billets with dimension of 20 mm × 20 mm × 45 mm. The rotary-die (RD) ECAP device [19,20] which had two crossed channels with a square section at 90° was adopted for ECAP. Before ECAP processing, the sample was firstly solution heat-treated (SHT) at 430 °C for 20 h and quenched in water to the room-temperature. Then it would be processed by ECAP for eight passes at 300 °C. After ECAP processing, the sample was quickly cooled in water. The sample obtained following the above procedure will be designated as SHT + E8p sample hereafter.

After ECAP, the processed sample was cut into pieces along the flow plane [21], as shown in Figure 1. Then these pieces would be rolled at room temperature, 100 °C, and 150 °C by HR01 hot rolling machine. The rolling direction (RD) coincided with the extrusion direction (*X*) and the transverse direction (TD) coincided with the vertical direction (*Y*), as shown in Figure 1. The rolling reduction of each rolling pass was ~10% and the roller speed was 20 mm/s. When the SHT + E8p samples were rolled at 100 °C or 150 °C, the samples were pre-heated to 100 °C or 150 °C at a furnace and held for 30 min before rolling. Between two rolling passes, they would be annealed at 100 °C or 150 °C for 10 min. The samples processed by the above procedures will be designated as SHT + E8p + RT rolling, SHT + E8p + 100 °C rolling, and SHT + E8p + 150 °C rolling hereafter. For comparison, the solution treated AZ91 alloy was also subjected to rolling alone at room temperature, 100 °C and 150 °C. The total rolling reductions of different samples were shown in Table 1. The total rolling reductions were almost the maximum reduction of each sample which could be caught without cracks. It could be seen that the total reduction increased with the rolling temperature, and the total rolling reductions of the SHT + ECAP samples were higher than that of the SHT samples except rolling at room temperature. Especially, when the SHT + ECAP sample was rolled at 150 °C, the total rolling reduction was as high as ~51.33%.

**Table 1.** Rolling reduction comparison of the solution heat treated (SHT) and SHT + equal channel angular pressing (ECAP) 8-p processed samples at different rolling temperature.

| Samples | Rolling Temperature | Total Reduction |
|---------|---------------------|-----------------|
| SHT | Room temperature (25 °C) | ~20.81% |
| SHT + E8P | | ~17.39% |
| SHT | 100 °C | ~18.35% |
| SHT + E8P | | ~20.72% |
| SHT | 150 °C | ~33.33% |
| SHT + E8P | | ~51.33% |

Vickers Hardness was measured at different treated states by applying a load of 100 g for 15 s. The tensile test was conducted at room temperature with a strain rate of $5 \times 10^{-4}\,\text{s}^{-1}$. Dog-bone shaped specimens cut paralleling to the flow plan in the ECAP processed billets or rolling plan in the rolled sheet were used for tensile testing, as demonstrated in Figure 1. The tensile specimen of the ECAP processed alloy was 6 mm in gage length, 2 mm in width, and 1.0 mm in thickness. The gage length and

width of the tensile specimens of the rolled alloy were also 6 mm and 2 mm, but the thickness depended on the final thickness of the rolled sample. Microstructures were examined by the optical microscopy (OM) and transmission electron microscopy (TEM). The observed plan was also the *X-Y* plane in the ECAP processed and rolled samples. Before OM observation, the samples were polished and etched with an acetic-picral solution (10 mL acetic acid, 6g picric acid, 10 mL water, 80 mL ethanol). The TEM samples were prepared by the jet polishing with a solution of 2% perchloric ethanol at −20 °C.

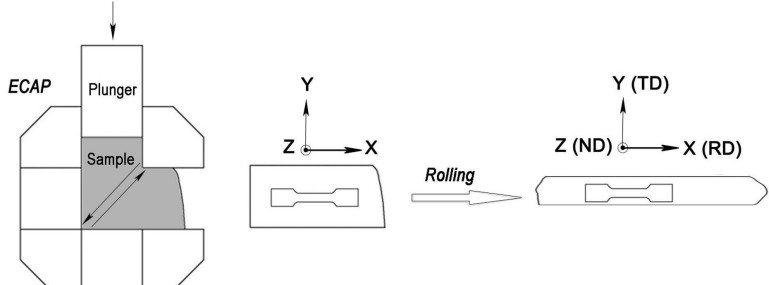

**Figure 1.** Schematic diagram of ECAP and the subsequent rolling processing, showing the ECAP flow plane (*X-Y*) and the rolling plane (RD-TD).

## 3. Results and Discussion

### 3.1. Microstructure

Figure 2a shows the optical microstructure of the as-cast AZ91magnesium alloy. It was a typical dendritic structure composed with $\alpha$ magnesium matrix, the divorced eutectic and some $\beta$-$Al_{12}Mg_{17}$ phase particles. After solution heat treatment, the eutectic structure and $\beta$-$Al_{12}Mg_{17}$ phase particles were almost dissolved in the matrix and the grain size grew up to ~150 μm. When the SHT sample was processed by ECAP for eight passes, precipitation happened simultaneously with the severe deformation. A great amount of super fine second phase particles spread uniformly in $\alpha$ magnesium matrix, as the black spots shown in Figure 2c. The average grain size was refined to <5 μm, and the size of the precipitated particle was only hundreds of nanometers.

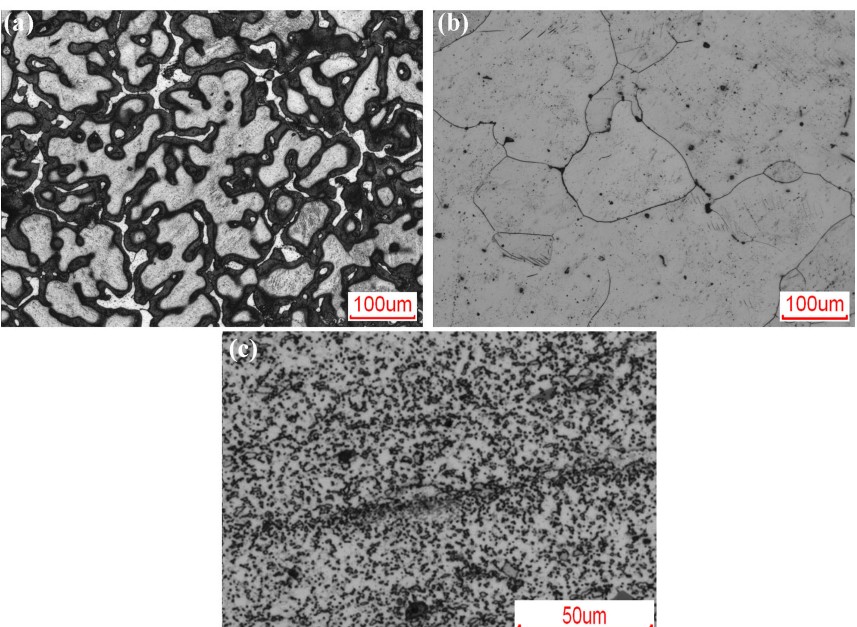

**Figure 2.** Optical micrographs of the AZ91 alloy: (**a**) as cast, (**b**) solution heat treaded, and (**c**) processed by solution heat treatment and eight passes of ECAP.

Figure 3a–c shows the microstructure of the SHT samples rolled at room temperature, 100 °C and 150 °C. Because the original grain size was big (~150 μm), and only limited slip system and twining were activated as the main deformation mechanisms at such low rolling temperature [22,23], the rolling formability of the SHT samples were poor. The highest rolling reduction was only reach to 33.33% of the SHT sample when it was rolled at 150 °C. As demonstrated in Figure 3a,b, when the SHT samples were rolled at room temperature and 100 °C, the grains in the observed rolled plan did not deform obviously and kept the original size (~150 μm). Twins can hardly be found. However, when rolling temperature was increased to 150 °C, the grains deformed clearly and a lot of twins could be seen inside the grains. The grains in all the rolled directly samples were not refined. It seemed that the inner strain induced by rolling at temperature 150 °C and lower were not high enough to activate the dynamic recrystallization. Figure 3c shows the twin density was very high in the sample rolled at 150 °C, which correspond to the increased formability of the alloy with the rolling temperature.

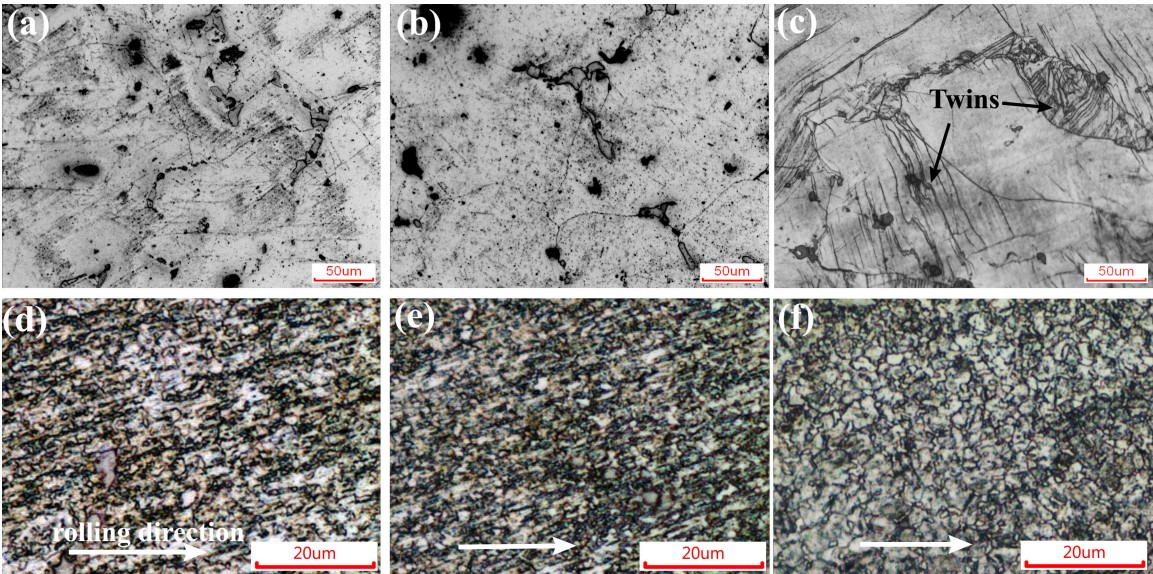

**Figure 3.** Optical micrographs of the AZ91 alloy in different processing state: (**a**) SHT + rolling at room temperature, (**b**) SHT + rolling at 100 °C, (**c**) SHT + rolling at 150 °C, (**d**) SHT + ECAP 8p + rolling at room temperature, (**e**) SHT + ECAP 8p + rolling at 100 °C, (**f**) SHT + ECAP 8p + rolling at 150 °C.

Figure 3d–f shows the microstructure of the SHT + E8p samples rolled at room temperature, 100 °C, and 150 °C. Both the SHT samples and SHT + E8p samples had a bad rolling formability at room temperature and 100 °C. The grains were greatly refined to <5 μm after processed by ECAP for eight passes. When the SHT + E8p samples were further rolled at room temperature and 100 °C, the microstructures were elongated obviously along the rolling direction, as shown as in Figure 3d,e. The grains could be seen elongated, but the average size of the grains and precipitates was not changed much that was very similar with that of the sample before rolling. More particles precipitated when the SHT + E8p sample was rolled at 100 °C (Figure 3e) than that rolled at room temperature (Figure 3d). When the SHT + E8p sample was rolled at 150 °C, as shown in Figure 3f, the microstructure became different from that rolled at room temperature and 100 °C. The grains were a little bigger but still very fine (~5 μm). The elongated structure cannot be seen clearly. Instead, due to the higher rolling and annealing temperature, recrystallization happened thoroughly and led the increased grain size. Many black second phase particles could also be seen inside the grains. The higher temperature also improved the rolling formability of the SHT + E8p sample. The rolling reduction was increased to 51.33%. Recrystallization happened during rolling and annealing for SHT + E8p sample, but which could not happen in the rolled alone sample.

Figure 4 shows the TEM microstructure of the SHT + E8p and SHT + E8p + 150 °C rolling samples. After solution and eight passes of ECAP, the grains of the alloy was refined to ~3 μm, and a large

amount of ellipsoid second phase particles, as white arrows pointing in Figure 4a,b, precipitated uniformly with an average size of ~300 nm. These superfine second phase particles precipitated dynamically during ECAP hindered the dislocation moving effectively [18]. Dislocations tangled up around these particles and became dislocation walls. And then some dislocation walls transformed into sub-grain boundaries. The density of dislocations inside the refined grains was relatively low. By the followed rolling at 150 °C, the grain boundaries became much clearer and the average grain size was a little bigger than that of the SHT + E8p sample. However, many finer recrystallized grains formed during the rolling process and annealing, as circled in Figure 4b. The fine ellipsoid second phase particles distributed along the fine recrystallized grains. However, in some big grains, high density of dislocations could be seen tangled around the particles in the grain, as the inset shows in Figure 4b.

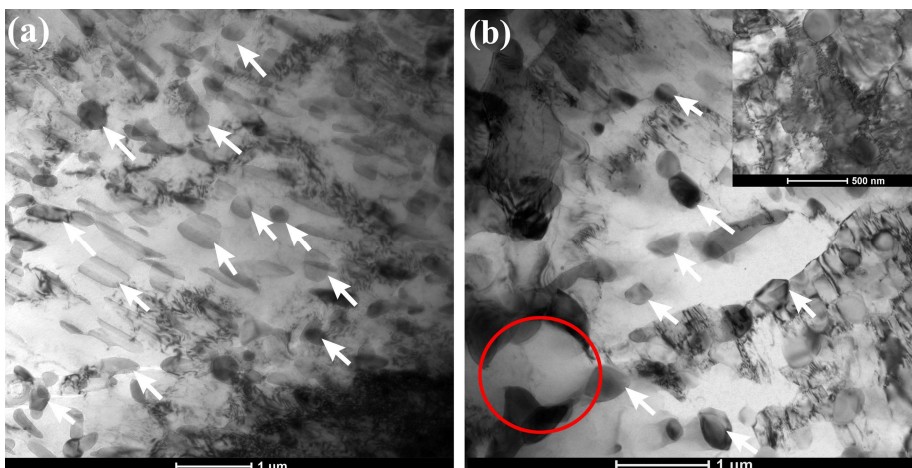

**Figure 4.** Transmission electron microscopy (TEM) micrographs of the AZ91alloy processed by (**a**) solution heat treatment and ECAP for eight passes, (**b**) SHT + ECAP 8p + rolling at 150 °C showing high density of dislocations around some particles.

### 3.2. Mechanical Properties

The hardness of the alloys at different processing states was shown in Figure 5a. The hardness of the SHT sample was 76 ± 5.3 HV. By rolling alone at room temperature, 100 °C and 150 °C, the hardness was increased from 76 ± 5.3 HV to 93 ± 7.1 HV, 108 ± 6.5 HV, and 106 ± 8.9 HV, respectively. By eight passes of ECAP, the hardness of the alloy was increased to 85HV ± 8.5, and it was further improved to 94 ± 8 HV, 111 ± 10.1 HV, and 108 ± 9.2 HV by the followed rolling at room temperature, 100 °C and 150 °C. Figure 5b exhibits typical tensile stress-strain curves of the SHT AZ91 alloy sample and the SHT + rolling samples. After solution heat treatment, the alloy was homogenized that most of the brittle eutectic structure and $\beta$-$Al_{12}Mg_{17}$ phases dissolved, so the SHT alloy had a good ductility up to ~15.8% ± 2.1%, but the yield strength was only ~125 ± 8 MPa. Although the hardness of the alloy was improved notably by rolling alone or by ECAP + rolling, due to the big original grains in the SHT samples and poor ductility at such low temperatures, the related tensile mechanical properties of the rolled alone samples were not good. After rolling at room temperature, 100 °C and 150 °C, the strength of all the samples were increased. The sample rolled at 100 °C yielded the highest strength of 219 ± 14 MPa, but all the rolled alone sample showed brittle fracture that the elongations to failure were less than 3%. After rolling at low temperature of 150 °C and less, the AZ91 alloy exhibited a real bad formability and ductility.

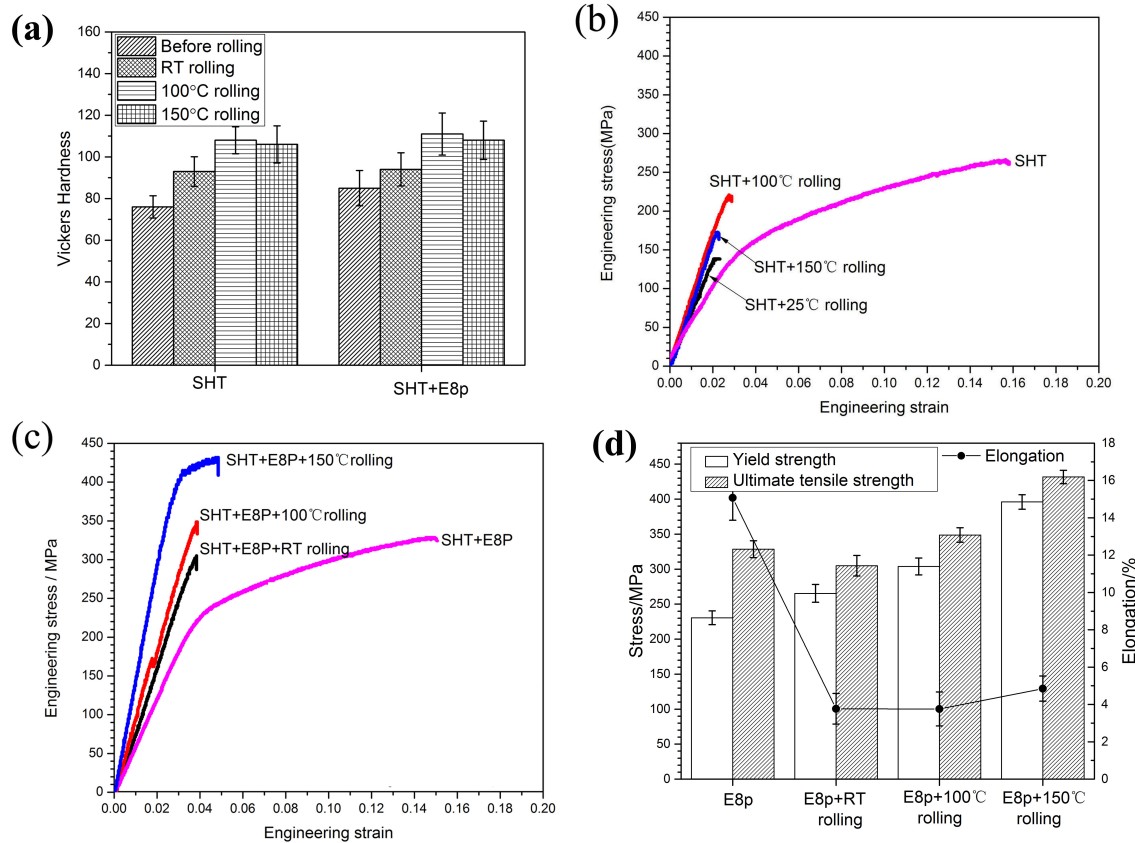

**Figure 5.** (**a**) The hardness comparison of the AZ91 alloy at different processing state, and typical engineering stress-strain curves of AZ91 alloys processed by (**b**) solution heat treatment (SHT), SHT + rolling at different temperature; (**c**) SHT + ECAP 8p, SHT + ECAP 8p + rolling at different temperature; and (**d**) tensile properties comparison of samples in (**c**).

Figure 5c exhibits typical tensile stress-strain curves of the SHT + E8p and SHT + E8p + rolling AZ91 alloy samples and Figure 5d shows the tensile properties comparison of these samples. When the solution heat treated sample was processed by eight passes of ECAP, different from the brittle fracture of the rolled alone samples, the strength was increased without loss in ductility. The yield strength and ultimate tensile strength were increased to 230 ± 9.8 MPa and 328 ± 10.3 MPa, respectively. More importantly, the ductility (~15.1% ± 1.2%) was not decreased after ECAP comparing to the SHT sample. The fine microstructure played an important role in these positive changes. By the followed rolling, the hardness of all the SHT + E8p samples was further increased. And correspondingly, an evident increase happened in the strengths. Especially, the yield strengths of the alloy were increased to 396 ± 10.3 MPa when it was processed by ECAP and rolled at 150 °C, which was hardly achieved in AZ91 alloys through any wrought procedure at high temperature. Because the rolling was conducted at lower temperature comparing to the conventional rolling, the tensile ductility of the SHT + E8p + rolling samples was decreased remarkably. By comparing the mechanical properties of the SHT + E8p + rolling samples, it can be found that the sample rolled at 150 °C had the best tensile mechanical properties. It yielded the highest tensile yield strength (396 ± 10.3 MPa) and ultimate strength (432 ± 9.5 MPa) in all the studied samples, and the elongation to failure (~4.8% ± 0.7%) was also acceptable and better than other rolled samples.

### 3.3. Discussion

Magnesium alloys were usually rolled or extruded at elevated temperature (>300 °C) because of their poor formability at ambient temperature. When the AZ91 alloy was rolled at ambient or low temperature (100 °C and 150 °C) alone at this work, the rolling formability was poor and the grains

cannot be refined. Only basal slip and twining mechanism could be activated in the magnesium alloys at temperature lower than 225 °C [22,24]. Therefore, when the SHT samples were rolled at 150 °C or lower temperatures, it was difficult to coordination the deformation with ambient grains which directly caused stress concentration and breakage in the alloy. All the tensile strengths and ductility of the rolled ZK60 alloy were very low. Although the hardness differences between the rolled samples were small, the tensile strengths were very low and showed much distinction. All the rolled samples failed in the elastic regime and showed brittle features that leads the un-coincide between strength and hardness. When the alloy was processed by ECAP at 300 °C, dynamic recrystallization was activated to refine the grains notably and many fine ellipsoid second phase particles precipitated uniformly during ECAP. Such homogeneous microstructure with fine grains and precipitates accounted for the good strength and ductility of the alloy.

By combining ECAP with low temperature rolling, the hardness and strength of the alloy were further increased, but the ductility decreased greatly. A large number of dislocations accumulated during the subsequent rolling procedure after ECAP. Especially, high content of Al element precipitated as fine ellipsoid particles ($\beta$-$Mg_{17}Al_{12}$) during ECAP which pinned up and hindered the dislocation movement effectively that also increased the dislocation density in the alloy. The increased dislocations and fine particles undoubtedly increased the work hardening of the alloy, but there was almost no room for more dislocations Therefore, the SHT + E8p + rolling samples had a super high strength, but sacrificed most of the ductility. These samples still failed in their elastic regime and showed brittle feature.

When rolled at 150 °C after ECAP, the SHT + E8p + rolling 150 °C sample had a super high strength that the yield strength and ultimate tensile were up to 396 ± 10.3 MPa and 432 ± 9.5 MPa, respectively, and an acceptable ductility of 4.8% ± 0.7%. The strength and ductility improved with the rolling temperature. Truly, the rolling temperature was important. The sample had the highest reduction at 150 °C that indicated the best rolling formability of the alloy. By former solution heat treatment and ECAP procedure, the sample had a homogeneous fine microstructure. The fine grains and increased grain boundaries enhanced the formation coordination. By increasing the rolling temperature to 150 °C, grain boundary slide might be activated in such fine structure that helped for the good formability of the alloy. Combining the higher rolling temperature and strain, recrystallization tended to happen partially during rolling and annealing at 150 °C, which could not found in the SHT sample when it was rolled at 150 °C. Furthermore, the presence of fine second phase particles also favored the recrystallization through the mechanism of particle stimulated nucleation (PSN). The ECAP processed AZ91 alloy had a lower recrystallization temperature comparing to the SHT alloy. The recrystallization released some severe strain and reduced the dislocation density in some grains, which mainly resulted in the moderate ductility of the alloy.

## 4. Conclusions

In conclusion, the high strength and moderate ductility could be achieved in AZ91 magnesium alloy by ECAP and the following low temperature (150 °C) rolling. The alloy had fine grains and good mechanical properties after processed by pre-solution heat treatment and ECAP. By the following rolling at 150 °C, the yield strength and ultimate strength were further improved up to 396 ± 10.3 MPa and 432 ± 9.5 MPa. Such high strengths were attributed to the fine grains and high density of dislocations. The uniformly distributed fine precipitates also supplied precipitate hardening that effectively hindered the dislocation movement. The good rolling formability and a moderate ductility could also be achieved when rolling at 150 °C. Recrystallization which happened during rolling and annealing was the main reason for the moderate of ductility.

**Author Contributions:** Y.Y., J.J. and A.M. designed the project and guided the research; Y.Y., Q.G., and Q.X. performed the experiment and analyzed the data; Y.Y wrote and revised the manuscript; H.L., J.S., Y.W. and D.S. reviewed the manuscript.

**Funding:** This research was funded by the Natural Science Foundation of China (Grant No. 51701065), the Fundamental Research Funds for the Central Universities (Grant No. 2017B01414), the Natural Science Foundation of Jiangsu Province (Grant No. BK20160867, BK20160869, and BK20180508), and the Natural Science Foundation of China (Grant No. 51774109).

**Conflicts of Interest:** The authors declare no conflict of interest.

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
