# Peer review of "High Mechanical Properties of AZ91 Mg Alloy Processed by Equal Channel Angular Pressing and Rolling"

_metals, doi:10.3390/met9040386_

Reviewer 1 Report

In their work. the authors studied the mechanical properties of a AZ91 alloy which was processed by Equal Channel Angular Pressing and Rolling. Although the results seem promising, there are several concerns with the manuscript in its current form.

Firstly, the language requires heavy editing and I would strongly urge the authors to take professional editing services to edit this manuscript.

Figures 3(d),(e) and (f) are not very clear and the authors need to use higher magnification images to reinforce their mechanism.

The authors show good mechanical properties in Figure 5(c). How many samples have they tested and what is the average and standard deviation of these properties? I strongly believe that the standard deviation of ductility in particular would be within each other's range.

The main issue with magnesium is limited ductility. Despite all the processing, the materials still exhibit very limited ductility of<5% which is not in the acceptable limits for a metal sheet to be used. How will the authors justify this?

How does the SHT+E8P+150C rolling show higher strength than SHT+E8P+100C rolling especially when there are more refined grains and higher number of dislocations in the latter? 

THe authors have no discussed the secondary phase morphology structure and distribution and their effects on the mechanical properties in the manuscript. This has a significant role to play in deciding the properties.

Overall, the manuscript lacks a coherent flow and does not present enough technical discussion on the results that the authors obtained.

Author Response

In their work. the authors studied the mechanical properties of a AZ91 alloy which was processed by Equal Channel Angular Pressing and Rolling. Although the results seem promising, there are several concerns with the manuscript in its current form.

Firstly, the language requires heavy editing and I would strongly urge the authors to take professional editing services to edit this manuscript.

Response: Thank you for your suggestion. We have carefully checked the manuscript to eliminate some bad wording and grammatical errors. In the revised manuscript, all changes and modifications are highlighted in red.

Figures 3(d),(e) and (f) are not very clear and the authors need to use higher magnification images to reinforce their mechanism.

Response:  We have changed Figures 3(d),(e) and (f) with higher magnification images. I hope that will help you and other readers to see more details in these images.

The authors show good mechanical properties in Figure 5(c). How many samples have they tested and what is the average and standard deviation of these properties? I strongly believe that the standard deviation of ductility in particular would be within each other's range.

Response:  We measured 10 points at each sample for microhardness test, and test at least 3 tensile samples to capture the reliable tensile properties. The standard deviations of these properties have been added in the modified Figure 5.

The main issue with magnesium is limited ductility. Despite all the processing, the materials still exhibit very limited ductility of<5% which is not in the acceptable limits for a metal sheet to be used. How will the authors justify this? 

Response: Truly the ductility or formability was the main issue for conventionally processed magnesium alloy. In our work, we achieved ~15.1% of ductility in AZ91 alloy by ECAP. Then we try to get higher strength by following low temperature rolling. It is not easy to control the rolling parameters at such low processing temperature, so when rolling at room temperature and 100 ˚C, the processed sample seemed very brittle and failed in the elastic regime.

However, hopefully when we rolled the SHT+E8p sample at higher temperature(150 ˚C) , we finally got an attractive high enough strength (~432MPa) for AZ91 alloy, and the lowest acceptable ductility (~4.8, very close to 5%). We know this properties were still not very good, but I think if we are keep doing some research to control the microstructure carefully, strength higher than 400 MPa with ductility >5% will be achieved in AZ magnesium alloys with no problem.

How does the SHT+E8P+150C rolling show higher strength than SHT+E8P+100C rolling especially when there are more refined grains and higher number of dislocations in the latter? 

Response:  Just as I said, when rolling at room temperature and 100 ˚C, the processed sample seemed very brittle and failed in the elastic regime. The formability of the SHT+E8P alloy was very poor, and dislocations accumulated but cannot recover well between two rolling passes. Therefore the deformation capability was exhausted and directly caused fracture early.   

 However, when rolling at 150 ˚C, the recrystallized fine grains (~5μm) were just a little bit bigger but the dislocation density was lower. Moderate density of dislocations in fine grains would be good for achieving high strength and developing ductility.

THe authors have no discussed the secondary phase morphology structure and distribution and their effects on the mechanical properties in the manuscript. This has a significant role to play in deciding the properties.

Response:  Thank you for reminding us the precipitate hardening effect. It really played significant role in deciding the properties. Since solution treatment was conducted before ECAP, a large amount of fine second phase particles precipitated during ECAP and even the following rolling process. We have discussed this dynamical precipitate hardening effect in our earlier papers (Materials Science and Engineering: A 2013, 588(0): 329-334; Journal of Alloys and Compounds,2014, 594, 182-188). Here we added some discussion about this precipitate hardening effect in page 5 in the modified manuscript which was highlighted in red.

Overall, the manuscript lacks a coherent flow and does not present enough technical discussion on the results that the authors obtained.

Reviewer 2 Report

The paper shows that combined SHT, ECAP and subsequent low temperature rolling can increase the tensile strength with moderate ductility. However there are a few points which must be addressed :

The authors have mentioned ZK60 alloys were also subjecting to rolling in this work for comparison with AZ91 alloy but have not provided any results with figures/tables related to it. This is a major problem in this manuscript, if something is done as experiment  for comparison it must be supported by figures / tables otherwise there is no point in mentioning ZK60 alloy in the manuscript.

In figure 3(c) the twins must be clearly marked. The authors can also do an EBSD for showing the type of twins present.

In figure 4(b) few of the recrystallized grains and precipitates can be seen but it would be better in some of those are clearly marked in the figure. Also do the precipitates show any Orientation relationship with the matrix?

In the discussion section it was mentioned in line 189 "When the alloy was processed by ECAP at 300˚C, dynamic recrystalliztion was activated to refine the grains notably and many fine ellipsoid second  phase particles precipitated uniformly during ECAP". What are the fine ellipsoid second phase particles should  be mentioned clearly.

There are small grammatical and spelling mistakes in the manuscript which should be carefully corrected.

Author Response

The paper shows that combined SHT, ECAP and subsequent low temperature rolling can increase the tensile strength with moderate ductility. However there are a few points which must be addressed :

The authors have mentioned ZK60 alloys were also subjecting to rolling in this work for comparison with AZ91 alloy but have not provided any results with figures/tables related to it. This is a major problem in this manuscript, if something is done as experiment  for comparison it must be supported by figures / tables otherwise there is no point in mentioning ZK60 alloy in the manuscript.

Response:  Thank you for your carefully reviewing. We found it was a writing error that ZK60 should be corrected to AZ91. We had changed it to AZ91 in the revised manuscript.

In figure 3(c) the twins must be clearly marked. The authors can also do an EBSD for showing the type of twins present.

Response:  We have marked the twins in the revised manuscript. Thank you for your suggestion.

In figure 4(b) few of the recrystallized grains and precipitates can be seen but it would be better in some of those are clearly marked in the figure. Also do the precipitates show any Orientation relationship with the matrix?

Response:  We have marked the ellipsoid precipitates and the recrystallized grain in Fig. 4 in the revised manuscript.

In the conventional aging treated AZ91 alloy, the lamer second phase precipitates lay in the basal plans in the matrix. But in our work the dynamically precipitated particles were ellipsoid which shows less orientation relation with the matrix.

In the discussion section it was mentioned in line 189 "When the alloy was processed by ECAP at 300˚C, dynamic recrystalliztion was activated to refine the grains notably and many fine ellipsoid second  phase particles precipitated uniformly during ECAP". What are the fine ellipsoid second phase particles should  be mentioned clearly.

Response:  We added some description and discussion about this kind of ellipsoid particles in page 5.

There are small grammatical and spelling mistakes in the manuscript which should be carefully corrected.

Reviewer 3 Report

Review Comments

The study is a contribution to the field and addresses important facts.

Line 13 / Line 37

Please do not call it …ultrahigh strength… …superplasticity…

high strength and moderate ductility, like written in the summary is fine!

Line 26

low density and high specific strength: high specific strength relates to the low density, please rephrase

Line 105

Figure 3a and b: I do not agree on the visible formation of twins - there are no twins presented… Figure 3c shows twins… with a better quality image at higher magnification twins could be distinguished from slip bands

Line 101 / 106 / 110

Please add the grain size values!

Line 121

It would be helpful to mark the rolling direction in the micrographs in Fig 3 d and e…

Line 145 / 146

Please change hardness information in 76HV ± … (please add the standard deviation!!!)

Line 151 / 156

Please change strength of ~125MPa in 125 MPa ± … (please add the standard deviation!!!)

Same counts for strength values in 167, 173, 178, 202, 218, 219

Line 160

Figure 5a needs error bars at the mean hardness values, so does the bar chart in Figure 5c,

Here I would recommend to have a Figure 5d and move the bar chart of strength and elongation from Figure 3c in an extra Figure!

Line 168 / 179

Please change elongation value ~15.1% in 15.1 % ± … (please add the standard deviation!!!)

Please change elongation value ~4.8% in 4.8 % ± … (please add the standard deviation!!!),

also in line 203

Line 95, 96, 150 and Line 196

you name the beta-phase γ-Al12Mg17 in lines 95, 96, 150 and once β-Al12Mg17 in line 196

…in the Magnesium community it is β-Mg17Al12 … please change!

Author Response

The study is a contribution to the field and addresses important facts.

Line 13 / Line 37

Please do not call it …ultrahigh strength… …superplasticity…

high strength and moderate ductility, like written in the summary is fine!

Response:  We have changed “ultrahigh” to “high”, and avoid using “superplasticity”.

Line 26

low density and high specific strength: high specific strength relates to the low density, please rephrase

Response:  We changed “low density and high specific strength” into “low density or high specific strength ”.

Line 105

Figure 3a and b: I do not agree on the visible formation of twins - there are no twins presented… Figure 3c shows twins… with a better quality image at higher magnification twins could be distinguished from slip bands

Response: We have corrected such unsuitable description about twins in Fig. 3(a) (b), and marked the twins in Fig.3(c) with a better quality image. See details in the revised manuscript.

Line 101 / 106 / 110

Please add the grain size values!

Response:  Yes, we added the grain size values as you suggested. See details in the revised manuscript.

Line 121

It would be helpful to mark the rolling direction in the micrographs in Fig 3 d and e…

Response:  Yes, we marked the rolling direction with arrows Fig. 3d,e,f as you suggested. See details in the revised manuscript.

Line 145 / 146

Please change hardness information in 76HV ± … (please add the standard deviation!!!)

Line 151 / 156

Please change strength of ~125MPa in 125 MPa ± … (please add the standard deviation!!!)

Same counts for strength values in 167, 173, 178, 202, 218, 219

Response:  Yes, we added the standard deviation in these mentioned hardness and strength values. See details in the revised manuscript.

Line 160

Figure 5a needs error bars at the mean hardness values, so does the bar chart in Figure 5c,

Here I would recommend to have a Figure 5d and move the bar chart of strength and elongation from Figure 3c in an extra Figure!

Response: We made an extra Figure as Fig. 5d to compare the tensile properties as you suggested. And we added error bars at the mean hardness value in Fig. 5a and so does the error bars at the tensile property values in Fig. 5d.

Line 168 / 179

Please change elongation value ~15.1% in 15.1 % ± … (please add the standard deviation!!!)

Please change elongation value ~4.8% in 4.8 % ± … (please add the standard deviation!!!),

also in line 203

Line 95, 96, 150 and Line 196

Response:  Yes, we added the standard deviation in these mentioned elongation values. See details in the revised manuscript.

you name the beta-phase γ-Al12Mg17 in lines 95, 96, 150 and once β-Al12Mg17 in line 196

…in the Magnesium community it is β-Mg17Al12 … please change!

Response:  We have changed all the “γ-Al12Mg17” into “β-Mg17Al12 ”. Thank you again for your useful comments.

Reviewer 4 Report

The authors propose subsequent rolling of samples subject to 8 passes by ECAP as way for improving the strength of AZ91 alloy. According to the results, the procedure is successful when rolling is carried out at 150°C for the SHS+E8p alloy. Only for this material, the strength is very high (exceeding 400 MPa). However, the rest of rolled SHS and SHS+E8p samples are brittle, failing in the elastic regime (this should be clearly stated in the text). However, there is no good correlation between hardness measurements and tensile data. Hardness measurements are small between the different rolled materials in spite of their different microstructures. The authors should analyze the reason of this behaviour on the basis of the main strengthening mechanisms (grain size, dislocation density in rolled materials, precipitates hardening or texture. At this point, texture developed during rolling could have a large influence in the mechanical strength of rolled materials. The authors should also consider in the discussion that strain achieved during rolling increases with increasing the rolling temperature, so recrystallization observed at 150°C could result from the combination of strain and temperature, also favoured by the presence of second-phase particles through the mechanism of particle stimulated  nucleation (PSN).

It is also surprising the prevalence of twinning when the material is rolled at 150°C. When temperature is raised, CRSS for the different slip systems decreases, so their activation should be favoured but instead significant contribution of twinning is found. What is the reason of such behaviour?

It is rare the brittle behaviour of SHS samples disregarding the rolling temperature. Is the low strength due to development of defects such microcracks/flaws during the rolling stage? In any case, rolled SHS alloys evidence a negligible effect of grain size on their strength (the grain size is over 100 μm). The coarse grain size should result in low strength, but hardness is superior to that of SHS+E8p material. This suggest that hardening effect is induced by the high density of dislocations and/or strong basal texture generated during rolling, A similar behaviour is found for rolled SHS+E8p samples. In the material rolled at 150°C, there is certain degree of recrystallization and some grain coarsening. Thus, compared with the SHS+E8p alloy the following microstructural features are found: i) Dislocation density is reduced, ii) the nature, size and volume fraction of second-phase particles hardly changes and iii) the grain size is slightly increased. This should induce softening rather than hardening. What is the origin of the high strength of the SHS+E8p alloy rolled at 150°C?

Once these points are addressed in the manuscript, it will be suitable for publishing.

Author Response

The authors propose subsequent rolling of samples subject to 8 passes by ECAP as way for improving the strength of AZ91 alloy. According to the results, the procedure is successful when rolling is carried out at 150°C for the SHS+E8p alloy. Only for this material, the strength is very high (exceeding 400 MPa). However, the rest of rolled SHS and SHS+E8p samples are brittle, failing in the elastic regime (this should be clearly stated in the text). However, there is no good correlation between hardness measurements and tensile data. Hardness measurements are small between the different rolled materials in spite of their different microstructures. The authors should analyze the reason of this behaviour on the basis of the main strengthening mechanisms (grain size, dislocation density in rolled materials, precipitates hardening or texture. At this point, texture developed during rolling could have a large influence in the mechanical strength of rolled materials. The authors should also consider in the discussion that strain achieved during rolling increases with increasing the rolling temperature, so recrystallization observed at 150°C could result from the combination of strain and temperature, also favoured by the presence of second-phase particles through the mechanism of particle stimulated  nucleation (PSN).

Response:  Thank you so much for your good and useful advises. We have added some discussions based on these advices to the Discussion section which had highlighted in red. See details in the revised manuscript.

It is also surprising the prevalence of twinning when the material is rolled at 150°C. When temperature is raised, CRSS for the different slip systems decreases, so their activation should be favoured but instead significant contribution of twinning is found. What is the reason of such behaviour?

Response:  In magnesium alloys, CRSS for non-basal slip system can only decrease obviously at temperature higher than ~225 °C. So the basal slip and twining still are the main deformation mechanism when rolling at 150°C.

It is rare the brittle behaviour of SHS samples disregarding the rolling temperature. Is the low strength due to development of defects such microcracks/flaws during the rolling stage? In any case, rolled SHS alloys evidence a negligible effect of grain size on their strength (the grain size is over 100 μm). The coarse grain size should result in low strength, but hardness is superior to that of SHS+E8p material. This suggest that hardening effect is induced by the high density of dislocations and/or strong basal texture generated during rolling, A similar behaviour is found for rolled SHS+E8p samples. In the material rolled at 150°C, there is certain degree of recrystallization and some grain coarsening. Thus, compared with the SHS+E8p alloy the following microstructural features are found: i) Dislocation density is reduced, ii) the nature, size and volume fraction of second-phase particles hardly changes and iii) the grain size is slightly increased. This should induce softening rather than hardening. What is the origin of the high strength of the SHS+E8p alloy rolled at 150°C?

Response:  Yes. All the rolled brittle samples failed in their elastic regime, except for the reason of high density of dislocations, some micro-defects such as micro-cracks/flaws show be the other important reason. Because all the samples in our work were rolled maximally without any visible macro-cracks, but some unvisible micro-cracks/flaws may have already exited and directly led to the fracture in the elastic regime.

In our work, we are also surprised by the overwhelming effect of temperature on the deformation mechanism and mechanical properties of magnesium alloys. Since the limited slip system in low temperature, the rolled samples in our work mostly showed inexplicable relationship between the grain sizes, hardness, strength… You can considered that there has great deformation coordinate problem when magnesium alloys were deformed at low temperature, so even the grain size of cold rolled magnesium alloy  is big, it cannot deformed easily. Then the hardness will be high but it is very brittle and had low strength.

When the SHT+E8p sample was rolled at 150°C, it had much higher strength than the SHT+E8p sample. You displayed 3 microstructure features of the rolled SHT+E8p sample.

        i.            In my opinion, the first one should be contrary. Although, recrystallization happened partially during the rolling process, the dislocation density was still higher than that in the SHT+E8p sample. The total rolling strain was very high, and the particles pinned up and hindered the dislocation movement effectively that also increased the dislocation density in the alloy.

      ii.            The nature, size and volume fraction of second-phase particles hardly changes, but it really play a role to hinder the dislocation movement and increase the dislocation density, thus strengthening the alloy.

    iii.            the grain size is slightly increased. Sure, it looks like that the grains increased slightly. But if you pay attention on the microstructure in Fig. 3f and Fig.4b, you will find many finer recrystallized grains formed in the grain boundaries near the triangle areas. It also played positive effect to the strength and ductility of the alloy.

Thank you again for your good advises and questions! 

Once these points are addressed in the manuscript, it will be suitable for publishing.

Round  2

Reviewer 1 Report

The revised manuscript can be accepted after making changes to the grammatical errors that are prevalent throughout the manuscript.

Reviewer 4 Report

The revised manuscript is acceptable for publishing